# Olive Leaf Extracts for a Green Synthesis of Silver-Functionalized Multi-Walled Carbon Nanotubes

**DOI:** 10.3390/jfb13040224

**Published:** 2022-11-07

**Authors:** Hassna Mohammed Alhajri, Sadeem Salih Aloqaili, Seham S. Alterary, Aljawharah Alqathama, Ashraf N. Abdalla, Rami M. Alzhrani, Bander S. Alotaibi, Hashem O. Alsaab

**Affiliations:** 1King Abdullah Institute for Nanotechnology, King Saud University, Riyadh 11451, Saudi Arabia; 2Chemistry Department, College of Science, King Saud University, Riyadh 11451, Saudi Arabia; 3KACST-Oxford Petrochemical Research Center (KOPRC), King Abdulaziz City for Science and Technology, Riyadh 11542, Saudi Arabia; 4Department of Pharmacognosy, Faculty of Pharmacy, Umm Al-Qura University, Makkah 21955, Saudi Arabia; 5Department of Pharmacology and Toxicology, Faculty of Pharmacy, Umm Al-Qura University, Makkah 21955, Saudi Arabia; 6Department of Pharmaceutics and Pharmaceutical Technology, College of Pharmacy, Taif University, Taif 21944, Saudi Arabia

**Keywords:** green biosynthesis, nanotechnology, silver nanoparticles, carbon nanotubes, *Olea europaea*, olive extracts

## Abstract

Green biosynthesis, one of the most dependable and cost-effective methods for producing carbon nanotubes, was used to synthesize nonhazardous silver-functionalized multi-walled carbon nanotubes (SFMWCNTs) successfully. It has been shown that the water-soluble organic materials present in the olive oil plant play a vital role in converting silver ions into silver nanoparticles (Ag-NPs). Olive-leaf extracts contain medicinal properties and combining these extracts with Ag-NPs is often a viable option for enhancing drug delivery; thus, this possibility was employed for in vitro treating cancer cells as a proof of concept. In this study, the green technique for preparing SFMWCNTs composites using plant extracts was followed. This process yielded various compounds, the most important of which were Hydroxytyrosol, Tyrosol, and Oleuropein. Subsequently, a thin film was fabricated from the extract, resulting in a natural polymer. The obtained nanomaterials have an absorption peak of 419 nm in their UV–Vis. spectra. SEM and EDS were also used to investigate the SFMWCNT nanocomposites’ morphology simultaneously. Moreover, the MTT assay was used to evaluate the ability of SFMWCNTs to suppress cancer cell viability on different cancer cell lines, MCF7 (human breast adenocarcinoma), HepG2 (human hepatocellular carcinoma), and SW620 (human colorectal cancer). Using varying doses of SFMWCNT resulted in the most significant cell viability inhibition, indicating the good sensitivity of SFMWCNTs for treating cancer cells. It was found that performing olive-leaf extraction at a low temperature in an ice bath leads to superior results, and the developed SFMWCNT nanocomposites could be potential treatment options for in vitro cancer cells.

## 1. Introduction

The current trend and approach by researchers in pharmaceutical biotechnology is to produce nanoparticles and nanomaterials that are both eco-friendly and cost-effective. The synthesis of eco-friendly metal nanoparticles (MNPs) is easier nowadays because of effective green chemistry technologies that were developed in the previous decade. Plants are ideal for making huge amounts of nanoparticles for biosynthesis. When in the case of microbes, the nanoparticles degraded rapidly, but in the case of plant-derived nanoparticles, they were far more stable and produced more quickly [1]. Furthermore, plants create a greater variety of nanoparticles, including in the form or size of nanoparticles. The result of many other works demonstrates that olive extract, which is produced in vitro, is capable of biosynthesizing silver nanoparticles (Ag-NPs). The combination of nanoparticles with a length between 1 and 100 nm can produce large surface-to-volume ratios [2]. The surface area to volume ratio increases, as well as the particles’ physical, chemical, and biological characteristics.

Carbon nanotubes (CNTs) are carbon nanomaterials closed at both ends with a small, thin, hollow, and concentric cylindrical structure; they were first introduced by Lijima in 1991 [3]. Carbon nanotubes are classified into single-walled carbon nanotubes (SWCNTs) and multi-walled carbon nanotubes (MWCNTs). Single-walled carbon nanotubes consist of a single layer of graphene wrapped in a seamless cylinder. Furthermore, MWCNTs consist of several layers of graphene wrapped to form concentric tubes. CNTs have been widely used for biocompatible nanoparticle production owing to their chemical stability, their ability to adsorb or conjugate with a broad range of therapeutic molecules (such as proteins, antibodies, DNA, enzymes, and drugs), and their functionality as drug-delivery vehicles [3,4].

Decorating CNTs with metal, metal oxide, or metal sulfide nanoparticles has attracted much attention owing to their unique catalysis as well as peculiar electrical, magnetic, thermal, and optical properties. In particular, CNTs decorated with silver nanoparticles (Ag-NPs) are cheap, eco-friendly, and have broad application prospects in the biomedical industry, thus increasing their wider recognition [5,6]. The unique features of each material may be combined, and the interactions between the two components may bring forth new properties in order to fully use these types of nanomaterials. As a result, carbon-based Ag composites have gotten a lot of attention and are now a hot topic in science [7]. CNTs may be filled with metal using several processes, including in situ filling during arc discharge development of the material, molten salt sorptive procedures, and other wet chemical approaches [8]. Since silver nanoparticles (Ag-NPs) exhibit unique electrical, optical, and biological characteristics, this new class of nanomaterials is growing quickly, such as the one we made in this paper. As Ag-NPs readily aggregate in aqueous solution, their lifespan in solution is usually limited [4]. Functionalized carbon nanotubes can attach nanoparticles to overcome this issue [9,10,11]. SFMWCNT nanohybrids were synthesized using a variety of chemical reagents, irradiations, and templates.

Nanomaterials, specifically particles (1–100 nm) with multiple organic or inorganic layers, are promising candidates to improve the efficiency of cancer treatment. Over the last decade, the ability of CNTs materials to treat cancer has been investigated for various cancer types, especially breast cancer [12,13,14,15]. They have been used for delivering chemotherapeutic drugs to a particular location. CNT have been used in many biomedical applications as many studies indicated [16,17,18]. Different approaches, including functionalization, as well as their critical functions in targeting distinct intracellular locations and tumor microenvironments, have been explored to learn more about the development of CNTs as prospective safe drug delivery vehicles in cancer treatment. Furthermore, CNT has seen a lot of recent progress in the field of cancer detection and therapy. As a result, there are several studies that summarize CNTs and their safety for medical applications, such as the paper by Tang, L. [18], who comprehensively introduce the theranostic applications of CNTs against many cancer types from the perspective of various therapeutic targets and emphasize the combination therapeutic modalities based on the physiochemical features of CNTs and compare it with other many reported literatures. The study concluded that CNT was a safe and effective system.

In the present study, we aim to combine a comfortable, safe, and cheap method as the water extraction of olive extracts and production of SFMWCNTs, which could be utilized for further investigation on anticancer activities as a single agent or combined with other modalities of treatment. The usage of olive-leaf has the added benefit that nanotechnology processing industries may make use of this plant. It is possible to employ SFMWCNTs nanoparticles produced in this work as anti-cancer agents, and also in the medical field for other diseases. Results indicated the ability of SFMWCNTs to suppress cancer cell expansion and spread was tested using the different cancer cell line. The current approach shows the ability of nanomaterials to enhance cancer growth inhibition and improves the SFMWCNTs selectivity toward cancer cells.

## 2. Materials and Methods

### 2.1. Materials

The fresh green leaves of olive (*Olea europaea* L.) used in this study were harvested and collected from the Al-Jouf area in Saudi Arabia. Taxonomy was identified in the Botanical Department, College of Science, King Saud University, as previously published [19]. AgNO_3_ (99.80% Silver nitrate) and carbon nanotubes (CNTs) were purchased from Sigma Aldrich Chemical Co., St. Louis, MO, USA.

### 2.2. Preparation of the Water Extract from Olive Leaves

The olives leaves were washed to remove any impurities and were then air-dried for two days in an open room environment under standard temperature and pressure conditions. Finally, they were ground into a fine powder. In order to obtain high-efficiency products, a cold extraction process at a temperature of 0 °C was adopted using 100 mL of distilled water as a solvent and a powder according to the ratio of 1:10. A schematic representation of the process is shown in Figure 1.

### 2.3. Preparation and Characterization of SFMWCNTs

#### 2.3.1. Preparation of SFMWCNTs Films

For preparing Ag functionalized MWCNTs (SFMWCNTs), 1 mL of the olive-leaf aqueous extract was mixed with 10 mL of the 2 mol/L silver nitrate (AgNO_3_) solution in a dark chamber at room temperature (20 °C). The formation of Ag-NPs in the solution was confirmed via the color change of the mixture from colorless to dark brown [20]. After addition of Ag-NPs, 15 mg of CNT powder was added and mixed for 1 h.

#### 2.3.2. Characterization of SFMWCNTs Films

The synthesized Ag-NPs optical properties were characterized via UV–Visible spectroscopy in the wavelength range of 200–700 nm. Then, JOEL JSM 7600F scanning electron microscopy (SEM) was used to study the morphology and know the size of the nanoparticles. The surface morphology of produced SFMWCNTs film was characterized by SEM at several magnifications after being placed on a glass substrate.

Average particles size and zeta potentials of biosynthesized silver NPs was determined by Dynamic Light Scattering (DLS) technique using a Beckman Coulter Delsa Nano C DLS Particle analyzer (Beckman Coulter, Inc., Fullerton, CA, USA) equipped with a 658-nm He-Ne laser. Polydispersity index (PI) values were also measured. Electron dispersive spectroscopy (EDS) was also studied to analyze the elemental compositions of the NPs (SEM-EDC). In addition, powder-XRD analysis was executed on an X-ray diffractometer (PAN analytical X-Pert PRO, UK) to determine the nanoparticles’ crystal density, purity, and size. Additionally, in order to identify the chemical responsible for the creation of Ag NPs, individual Fourier Transform Infrared spectroscopy (FTIR) measurements were performed for each procedure. Spectra were collected in the range of 400 to 4000 cm^−1^ and analyzed by removing the spectrum of pure KBr (potassium bromide).

### 2.4. Cell Culture

Three cancer cell lines, MCF7 (human breast adenocarcinoma, ATCC-HTB22), HepG2 (human hepatocellular carcinoma, ATCC HB-8065), and SW620 (human colorectal cancer, ATCC- CCL-227), were tested in this study. Two of the cell lines (MCF7 and SW620) were maintained in Roswell Park Memorial Institute Medium (RPMI-1640, Gibco, Life Technologies, Carlsbad, CA, USA), while the HepG2 cell line was cultured Dulbecco’s Modified Eagle Media (DMEM, Gibco, Life Technologies, Carlsbad, CA, USA). All of the cell lines were maintained at 37 °C in 5% CO_2_ and 100% relative humidity. All media were supplemented with 10% heat-inactivated fetal bovine serum (FBS, Gibco) and 1% penicillin–streptomycin antibiotic, consisting of 10,000 units of penicillin and 10,000 µg of streptomycin (Gibco) per mL.

### 2.5. Determination of Cytotoxicity Using MTT Assay

The cytotoxicity of the extracts/formulation (The SFMWCNTs) was evaluated by MTT assay, as previously reported [21]. The three cell lines were cultured separately in 96-well plates (3 × 10^3^ cells/well) and incubated at 37 °C overnight. The concentrations tested were 1, 50, 100, 500, 1000, and 5000 μM. Plates were incubated for different time points (24, 48, and 72 h), after which MTT was added to each well, and the plates were incubated for a further 3 h. The supernatant was removed, and the MTT product was solubilized by adding DMSO to each well. Absorbance was read using a multi-plate reader (BIORAD, PR 4100, Hercules, CA, USA). The optical density of the purple formazan A550 was proportional to the number of viable cells. Doxorubicin (range between 0.001–10 μM) was used as a positive control, and IC_50_ values were determined using GraphPad Prism (San Diego, CA, USA), as listed in Table 1.

## 3. Results

### 3.1. Preparation and Characterization of SFMWCNTs

As shown in Figure 2A,B, the tubes showed the formation of Ag-NPs in the mixture was confirmed via the change in color from yellowish to dark brown. This change occurs due to the reduction of Ag ions to nanosized Ag particles, which in turn is caused by the water-soluble organic materials present in the plant [22].

The UV–Vis absorption spectra of the SFMWCNTs obtained by mixing the olive-leaf extract with the Ag-NPs is shown in Figure 2C. The peak at 419 nm indicates the change in color to dark brown, thus confirming the formation of silver NPs. By contrast, the UV–Vis spectra of the Ag/SFMWCNT nanocomposite exhibit an absorption peak at around 430 nm. Additionally, the particle size for the prepared nanocomposite was 257 nm, and the zeta potential was −24 mV, indicating its compatibility to be used for systemic treatment for further in vivo testing.

Furthermore, Figure 3 shows the XRD patterns for the olive-leaf extract and the synthesized SFMWCNTs. The peaks for the SFMWCNTs are comparable with those of the standard card (JCPDS card No. 89-3722). In particular the 2θ diffraction peaks at 38.09°, 44.17°, 64.49°, and 77.06° can be attributed to the (111), (200), (220), and (311) planes of Ag, respectively. This figure indicates that the Ag-NPs have a crystalline nature, and the low intensity of the Ag peaks shows that only a small amount of Ag has been used.

### 3.2. Fourier Transform Infrared Spectroscopy (FTIR) Results

FTIR spectroscopy was used to discriminate and identify the biomolecules of olive leaf. FTIR has been utilized by numerous researchers to analyze a variety of materials [23,24,25]. By looking at FTIR spectra based on stretching or bending vibration of specibonds, FTIR spectroscopy can reveal information on intermolecular interaction [25,26]. In Figure 4, the olive leaf’s FTIR spectrum is shown. An intense broad band was seen at 3384.58 cm^−1^, which was caused by polyphenols O-H stretching modes. Phenolic compounds and alcohols both contain the hydroxyl (OH) group, which has a broad absorption band at 3310.7 cm^−1^. The C=C stretch vibration in the aromatic ring and the C=O stretch vibration in polyphenols may have also been responsible for another strong band at 1612.02 and 1386.23 cm^−1^. The C-H and O-H stretches in alkanes and carboxylic acids have been observed to surface at 2935.37 cm^−1^, respectively. The C-O bond stretching in amino acids has also resulted in the emergence of a band at 1078.55 cm^−1^. Previous research has shown that the O-H/N-H, C=C, and C-O-C stretching vibrations are responsible for the FTIR bands that developed at 3384.58 cm^−1^, 1612.02 and 1386.23 cm^−1^, and 1078.55 cm^−1^ [23,24].

### 3.3. Morphological Studies: SEM and EDS Measurements

The SEM and EDS images of the olive-leaf extract and SFMWCNT nanocomposite prepared by green reduction method are shown in Figure 5 and Figure 6, respectively. SEM images were used to study the morphology of nanomaterials, as shown in Figure 5. The synthesized material exhibits clusters with almost a circular shape, whereas the nanocomposite has a tubular-channel structure. Thus, these images indicate amorphous and crystalline structures, respectively. The current findings are compatible with our XRD results.

In addition, based on our previous protocol for the extracted spherical olive leaf and SFMWCNT nanocomposite, the presence of Ag-NPs on the SFMWCNT sheets was also elementally identified via EDS analysis (Figure 6). As shown, the weight percent of silver in the composite is 65.33%, whereas no silver element is present in the extract [20].

### 3.4. Cytotoxicity Effects of SFMWCNT

During this study, the cytotoxicity of the extracts/formulation (SFMWCNTs) was evaluated by MTT assay. The study was conducted to determine the SFMWCNT ability to kill the cancer cells, MCF7 (human breast adenocarcinoma), HepG2 (human hepatocellular carcinoma), and SW620 (human colorectal cancer) after 24, 48, and 72 h. F1: plant extract; F2: CNTs (without drug); F3: Ag/SFMWCNTs with the extract; Dox: Doxorubicin (a chemotherapeutic agent) as shown in Figure 7, Figure 8 and Figure 9 (A, B, and C). The MTT cytotoxicity study identified the IC_50_ of the three cell lines after treatment with a wide range of concentrations from SFMWCNT (1–5000 µM), as shown in Table 1.

The IC_50_ for MCF7 was found to be 169.35 μM for plant extract (F1) compared to 15.78 μM for Ag/SFMWCNTs (F3) after 24 h and a similar significance difference was clearly noticed with 48 h and 72 h, respectively, as shown in Figure 7 and Table 1. Figure 8 shows higher sensitivity for SFMWCNTs (F3) against HepG2 compared with the extract and control, especially after 72 hr as the IC_50_ values were 69.49, 54.27, and 1.85 μM for (F1) plant extract; (F2) CNTs; (F3) SFMWCNTs, respectively. Interestingly, the most cytotoxic activity upon treatment with the SFMWCNTs was noticed against SW620 cells as shown in Figure 9 and Table 1. It is indicated that SFMWCNTs have more effect on human colorectal cancer with lower IC_50_ values which were 5.80, 4.97, and 0.49 μM for 24 h, 48 h, and 72 h, respectively. The cytotoxic effect against all three cell lines was prominent with low IC_50_ values, especially after 72 h. It could indicate greater release with time for SFMWCNTs and has shown promise as a therapy for cancer cells. This treatment may result in the strongest suppression of cancer cells.

## 4. Discussion

Green silver nanoparticle production using plant extracts as reducing agents has been reported in recent studies [27,28,29]. Other biological techniques have several benefits, including being cost-effective, simple to use, and environmentally friendly. One-step metal ion reduction using biomolecules found in plant extracts results in nanoparticles [30,31]. Chemical and physical techniques have been substituted by biosynthesis of nanoparticles as a cost-effective and environmentally acceptable alternative [32]. Many secondary metabolites and biomolecules are found in natural plant extracts, including flavonoids, alkaloids, terpenoids, phenolic compounds, and enzymes. Flavonoids, for example, are found in abundance. Metal ions can be reduced to NPs using secondary metabolites in one-step synthesis methods that are favorable to the environment [33]. Green synthesis generally eliminates the need for stabilizing and capping agents and produces biologically active molecules that are shape- and size-dependent, which reduces the need for additional chemicals [34]. In our study, reduction of aqueous Ag+ with olive extracts at room temperature yielded Ag-NPs. Electrochemical, chemical, and physical techniques have been used to manufacture Ag-NPs coated on MWCNTs. In another study, water extract of Satureja hortensis L was used as a reducing and stabilizing ingredient to green synthesize and characterize the Ag/ FMWCNT nanocomposite at room temperature [35].

In previous works, it has been demonstrated that functionalizing CNTs is a viable strategy to enhance breast cancer treatment’s biocompatibility and cytotoxicity effect [36]. The green synthesis represents a more reliable and price-efficient approach compared with other decoration techniques. Thus, in this study, the green technique reproduced from Pirtarighat et al. [37] for preparing Ag-NPs MWCNTs composites using plant extracts was followed [38]. As for the extraction process, the olive leaves were dried and ground before being extracted at 0 °C with distilled water. This process yielded various compounds, the most important of which were Hydroxytyrosol, Tyrosol, and Oleuropein [39,40]. Subsequently, a thin film was fabricated from the extract, resulting in a natural polymer [40]. Olive-leaf extracts contain medicinal properties and combining these extracts with Ag-NPs is often a viable option for targeted drug delivery [41]; thus, this possibility was employed for cancer cells [42,43].

Silver nanoparticles derived from olive leaf extract demonstrated antitumor action in our earlier studies [33] and were similar to our results [44,45]. It is possible to study the effect of nanoparticles in vivo in the future to test their feasibility to be used in the development of new nanomedicines. Polyphenol concentration in olive leaf extract was high, indicating that it has powerful antitumorigenic effects as previously reported [46]. A natural anticancer drug may be made using an environmentally friendly process such as Ag-NP green synthesis, as proven by the findings in this paper. Ag-NP green synthesis has been shown to be effective on different cancer cell line types, as demonstrated in the current study’s data. More research and therapeutic applications were in agreement with our findings, which indicates it might be beneficial to present them as anticancer treatment options [47,48,49].

## 5. Conclusions

Sustainable methods for synthesizing metallic nanoparticles are essential in the field of nanotechnology. There is a growing consensus that nanoparticles are the cornerstones of nanotechnology. Due to their appealing physiochemical characteristics, silver nanoparticles play an important role in biology and medicine. In the present study, it was found that performing olive-leaf extraction at a low temperature in an ice bath leads to superior results, as most organic compounds are destroyed at high temperatures. The olive-leaf extract contains a large quantity of organic compounds, which have several applications in medicine, condensates, and solar cells. Furthermore, it was found that when these organic compounds are linked to CNTs, the efficiency is hugely increased. Toxic solvents and waste are avoided by using olive leaf extracts as a natural, low-cost biological reducer to create metal nanostructures using an efficient green nanochemistry approach. The obtained materials were analyzed using Ultraviolet–Visible spectroscopy (UV–Vis), scanning electron microscopy (SEM), electron dispersive spectroscopy (EDS), and powder XRD. These findings demonstrate a simple, quick, and cost-effective way to make silver nanoparticles. The usage of olive-leaf has the added benefit that nanotechnology processing industries may make use of this plant. It is possible to employ SFMWCNTs nanoparticles produced in this work as anti-cancer agents, and also in the medical field for other diseases.

## Figures and Tables

**Figure 1 jfb-13-00224-f001:**
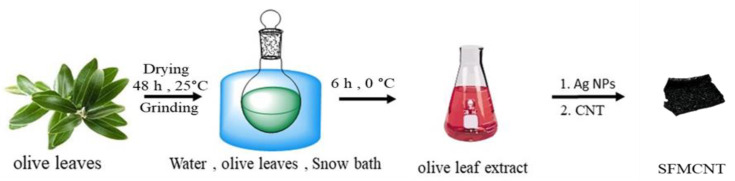
Preparation of the water extract from olive leaves and formulation process of nonhazardous silver-functionalized multi-walled carbon nanotubes (SFMWCNTs).

**Figure 2 jfb-13-00224-f002:**
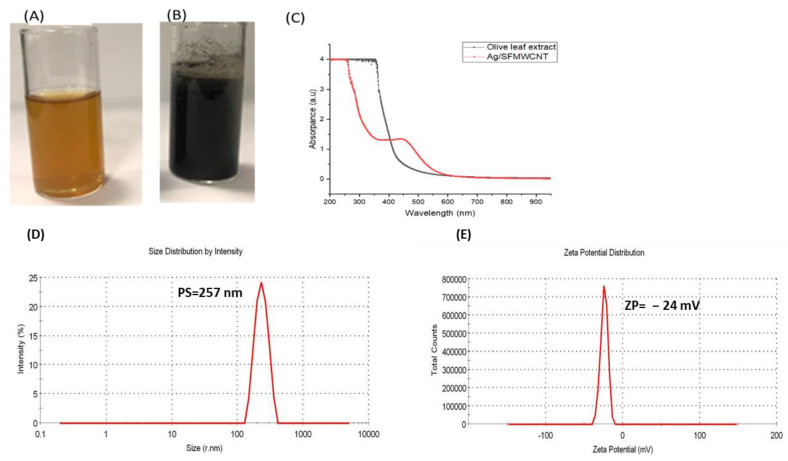
(**A**) Olive-leaf extract. (**B**) SFMWCNTs with olive-leaf extract. (**C**) UV–Vis. absorption spectra of olive-leaf extract and Ag/SFMWCNT with olive-leaf extract. (**D**) Particle size (PS) of SFMWCNTs. (**E**) Zeta potential (ZP) of SFMWCNTs.

**Figure 3 jfb-13-00224-f003:**
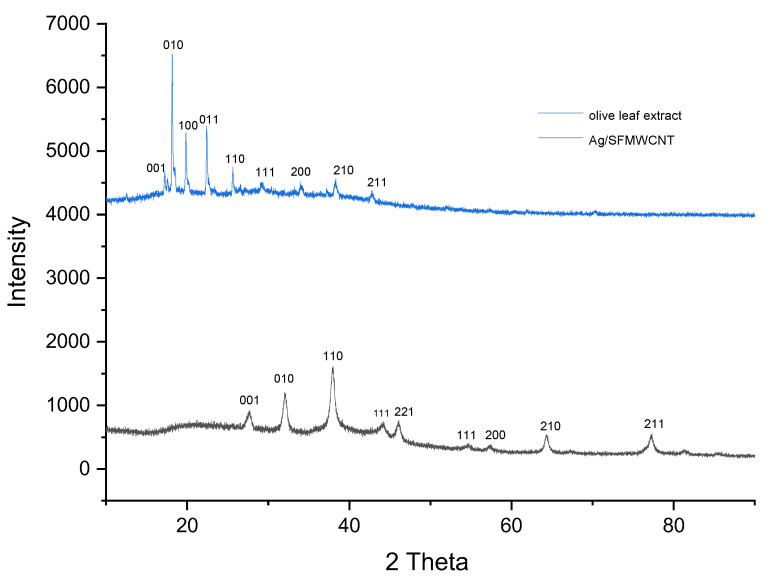
XRD patterns of the olive-leaf extract and XRD patterns of the Ag/SFMWCNTs nanocomposite were synthesized using the olive-leaf extract after 1 h from fabrication.

**Figure 4 jfb-13-00224-f004:**
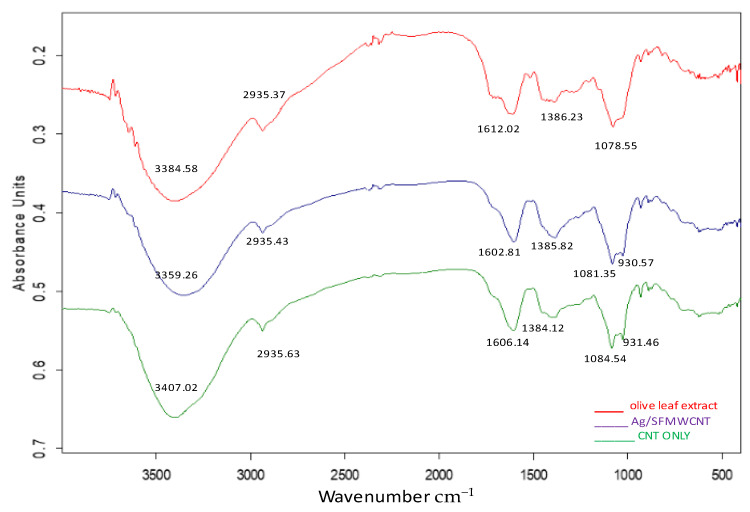
FTIR spectroscopy results of the olive-leaf extract and Ag/SFMWCNTs nanocomposite. It can be seen from the FTIR spectrum of silver NPs that the wavenumber of the OH bending vibration at 3359.29 cm^−1^ appeared to broaden and decrease, and that the peaks in the (1700–400) cm^−1^ region almost underwent alteration.

**Figure 5 jfb-13-00224-f005:**
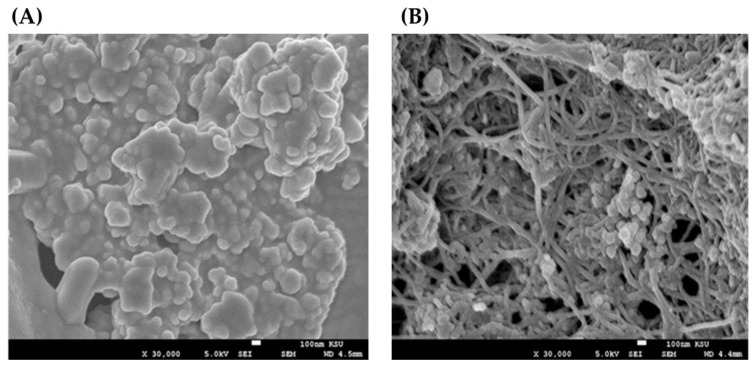
SEM images of (**A**) the prepared olive-leaf extract and (**B**) synthesized Ag/SFMWCNT nanocomposite. The synthesized material exhibits clusters with almost a circular shape, whereas the nanocomposite has a tubular-channel structure. Thus, these images indicate amorphous and crystalline structures, respectively.

**Figure 6 jfb-13-00224-f006:**
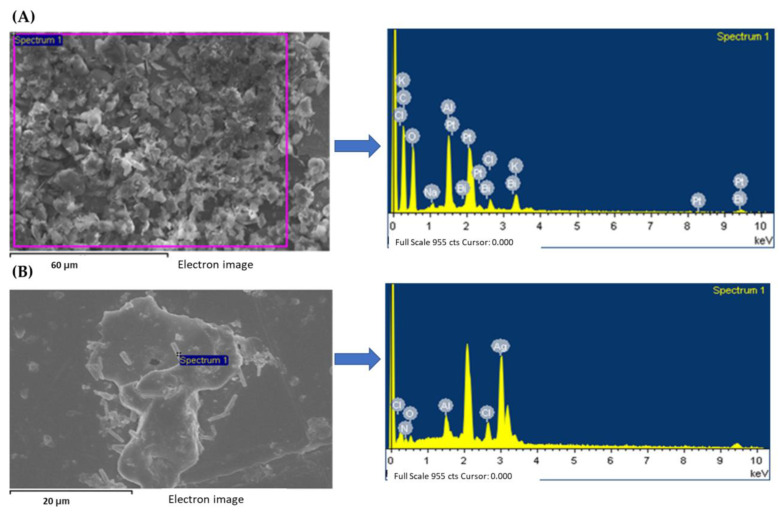
(**A**) EDS images of the olive-leaf extract. (**B**) EDS images of the Ag/SFMWCNT nanocomposite. The presence of Ag-NPs on the SFMWCNT sheets was also elementally identified via EDS analysis. As shown, the weight percent of silver in the composite is 65.33%, whereas no silver element is present in the extract.

**Figure 7 jfb-13-00224-f007:**
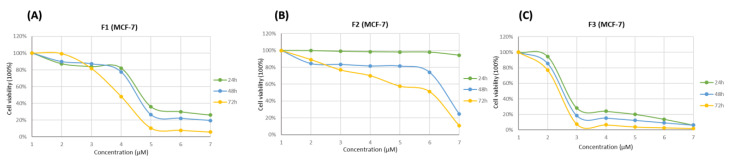
Cytotoxicity of SFMWCNTs on MCF7 cancer cell lines (human breast adenocarcinoma) after 24, 48, and 72 h. (**A**) F1: plant extract; (**B**) F2: CNTs (without drug); (**C**) F3: Ag/SFMWCNTs with the extract; Dox: Doxorubicin (chemotherapeutic agent).

**Figure 8 jfb-13-00224-f008:**
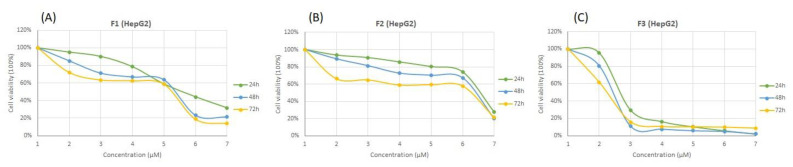
Cytotoxicity of SFMWCNTs on HepG2 cancer cell lines (human hepatocellular carcinoma) after 24, 48, and 72 h. (**A**) F1: plant extract; (**B**) F2: CNTs (without drug); (**C**) F3: Ag/SFMWCNTs with the extract; Dox: Doxorubicin (chemotherapeutic agent).

**Figure 9 jfb-13-00224-f009:**
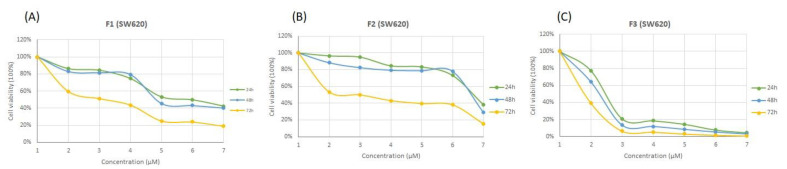
Cytotoxicity of SFMWCNTs on SW620 cancer cell lines (human colorectal cancer) after 24, 48, and 72 h. (**A**) F1: plant extract; (**B**) F2: CNTs (without drug); (**C**) F3: Ag/SFMWCNTs with the extract; Dox: Doxorubicin (chemotherapeutic agent).

**Table 1 jfb-13-00224-t001:** Cytotoxicity of the studied extracts and formulation against three cancer cell lines (MTT 24, 48, 72 h, IC_50_ ± SD μM).

Sample	MCF7			HepG2			SW620		
	24 h	48 h	72 h	24 h	48 h	72 h	24 h	48 h	72 h
F1	169.35 ± 19	151.20 ± 18	93.97 ± 2.29	261.2 ± 26	142.55 ± 14	69.49 ± 7.55	126.30 ± 26	121.25 ± 15	2.74 ± 0.24
F2	1003.35 ± 126	1119 ± 103	375.10 ± 9.05	1091 ± 70	600.9 ± 15	54.27 ± 4.20	958.90 ± 162	922 ± 147	3.86 ± 0.04
F3	15.78 ± 0.45	7.04 ± 2.62	2.94 ± 0.54	18.75 ± 4.87	4.51 ± 0.88	1.85 ± 0.09	5.80 ± 0.49	4.97 ± 1.31	0.49 ± 0.01
Dox	0.11 ± 0.01	0.10 ± 0.01	0.09 ± 0.01	0.63 ± 0.07	0.26 ± 0.02	0.11 ± 0.02	0.13 ± 0.01	0.10 ± 0.01	0.08 ± 0.01

F1: plant extract; F2: CNTs (without drug); F3: Ag/SFMWCNTs with the extract; Dox: Doxyrubicin (chemotherapeutic agent).

## Data Availability

Not applicable.

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
