# Peer review of "Olive Leaf Extracts for a Green Synthesis of Silver-Functionalized Multi-Walled Carbon Nanotubes"

_jfb, 2022, doi:10.3390/jfb13040224_

Round 1

Reviewer 1 Report

Green biosynthesis was used to synthesize nonhazardous silver-functionalized multiwalled carbon nanotubes (SFMWCNTs) successfully. It was found that performing olive-leaf extraction at a low temperature in an ice bath leads to superior results, and the developed SFMWCNT nanocomposites could be potential treatment options for in vitro cancer cells. The manuscript contains useful results, however, before publication some minor formal and scientific modifications should be made.

1. The Reference list should be checked. There are some trouble around [13].

2. English spell check should be done.

3. The CNTs are very effective for the stabilization of metal particles. Similarly the titanate nanotubes are also suitable support for metals including silver (Surface Science Reports, 71 2016, 473-546). Please make a short comparison or mention it in the Introduction.

4. Please try to calculate the size of the silver from XRD using Scherer equation)

5. Please give an estimation for the oxidation state of silver. Is this ionic or metallic? 

Author Response

Dear/ Respected reviewers

Thank you for giving us the opportunity to submit a revised draft of our manuscript titled [Olive Leaf Extracts for a Green Synthesis of Silver-Functionalized Multi-Walled Carbon Nanotubes], manuscript ID: jfb-1944808 to Journal of Functional Biomaterials. I and my co-authors appreciate the time and effort that you and the reviewers have dedicated to providing your valuable and positive feedback on our manuscript. We are grateful to the reviewers for their insightful comments on our paper. We tried as much as possible to respond to most of the enquiries and suggestions provided by the respected reviewers. All changes were made through Microsoft word track changes. Here is our point-by-point response to the reviewers’ comments and concerns followed by references to some responses.

Accept our regards,

Reviewer 1:

Green biosynthesis was used to synthesize nonhazardous silver-functionalized multiwalled carbon nanotubes (SFMWCNTs) successfully. It was found that performing olive-leaf extraction at a low temperature in an ice bath leads to superior results, and the developed SFMWCNT nanocomposites could be potential treatment options for in vitro cancer cells. The manuscript contains useful results, however, before publication some minor formal and scientific modifications should be made.

  1. The Reference list should be checked. There are some trouble around [13].

Answer:  we would like to thank the reviewer for noticing this. We have checked the mentioned reference. It was actually about the different applications of CNT for different types and there were 4 references were not updated by Endnote software.  We have modified the sentences with references as following:

Nanomaterials, specifically particles (1–100 nm) with multiple organic or inorganic layers, are promising candidates to improve the efficiency of cancer treatment. Over the last decade, the ability of CNTs materials to treat cancer has been investigated for various cancer types, especially breast cancer [1-4]. “

  1. English spell check should be done.

Answer: The manuscript has been revised for English language clarity and modifying all grammatical errors.

  1. The CNTs are very effective for the stabilization of metal particles. Similarly, the titanate nanotubes are also suitable support for metals including silver (Surface Science Reports, 71 2016, 473-546). Please make a short comparison or mention it in the Introduction.

Answer: We would like to thank the reviewer for the suggestion. We have added the following reference [5] as suggested and modified introduction accordingly.

  1. Please try to calculate the size of the silver from XRD using Scherer equation)

Answer: We have calculated the size from XRD using Scherer equation as the following:

peak position (2Ï´)

FWHM (β)

SIZE (nm)

27.71205

74.69306

0.109504219

27.7148

7301.58666

0.001120202

37.78432

0.01407

596.5421046

37.78432

36.95202

0.227141775

47.25113

5.04126

1.71934908

47.82766

32.29972

0.268947014

57.28496

0.45281

19.98294392

61.79424

2.37354

3.89902575

87.17371

105.05205

0.104362584

D=k λ/ β cos θ                   *k= 0.9             *λ=0.154 nm

AVERAGE SIZE (nm) = 69.20605546

  1. Please give an estimation for the oxidation state of silver. Is this ionic or metallic? 

Answer: It is ionic because Ag+ ( AgNO3 -----à Ag+ + NO3- )

Reviewer 2 Report

Comments:

1. Did author study the stability of the nanoparticles over the time?

2. Did author study the UV-Vis spectra from the effects of pH, function of temperature, varies concentration?

3. Many times author mentioned AgNO3 NPs and Ag-NPs. What are the difference between two? Its really confusing for the people to read.

4. Can author tell how many CNTs attached per Ag-NP?

5. Did author study the cellular uptake?

6.Did author try weight loss experiment nanoparticle due to desorption of organic moieties by TGA?

7. Did author see any changes in FT-IR spectrum for olive oil extract and formed nanoparticles?

8. Author should go through the from the line 110 to 113 and rewrite it. It is bit confusing to understand.

Author Response

Dear/ Respected reviewers

Thank you for giving us the opportunity to submit a revised draft of our manuscript titled [Olive Leaf Extracts for a Green Synthesis of Silver-Functionalized Multi-Walled Carbon Nanotubes], manuscript ID: jfb-1944808 to Journal of Functional Biomaterials. I and my co-authors appreciate the time and effort that you and the reviewers have dedicated to providing your valuable and positive feedback on our manuscript. We are grateful to the reviewers for their insightful comments on our paper. We tried as much as possible to respond to most of the enquiries and suggestions provided by the respected reviewers. All changes were made through Microsoft word track changes. Here is our point-by-point response to the reviewers’ comments and concerns followed by references to some responses.

Accept our regards.

Reviewer 2: Comments and Suggestions for Authors

Comments:

  1. Did author study the stability of the nanoparticles over the time?

Answer: Yes, we have conducted stability test at 2 and 4 months after SFMWCNT have been made for only nanoparticle size. We have noticed that NPs were stable and the increase in PS diameter was not significant, which indicates no aggregation happened during storage of the synthesized nanomaterials.

  1. Did author study the UV-Vis spectra from the effects of pH, function of temperature, varies concentration?
    Answer: We have not conducted this study as we have used previously validated method to make fictionalizations of CNT.
  2. Many times author mentioned AgNO3 NPs and Ag-NPs. What are the difference between two? Its really confusing for the people to read.
    Answer: Thank you for the comment. The change has been made as suggested. AgNo3 has been used for preparation part and the name of synthesized is silver or Ag-NPs.

“For preparing Ag functionalized MWCNTs (SFMWCNTs), 1 mL of the olive-leaf aqueous extract was mixed with 10 mL of the 2 mol/L silver nitrate (AgNO3) solution in a dark chamber at room temperature (20 °C). The formation of silver NPs in the solution was confirmed via the color change of the mixture from colorless to dark brown [6].”

  1. Can author tell how many CNTs attached per Ag-NP?

Answer: Based on the performed synthesis steps, we don’t know exactly the number of CNT attached per Ag-Np. We have taken in our consideration a previous report by Maleszewski, A. and his group [7]. The authors have stated that, “It should be noted that a calculation of the amount of silver present based only on the ppi and diametric distributions is a considerable underestimate for two main reasons. One reason is that nanoparticles smaller than 1 nm cannot be measured via traditional SEM. There are many particles which are observed in both the 80 °C and 60 °C reduced samples which are near this size, suggesting that a significant number of these very small particles may go uncounted. The second cause of error is due to shadowing effects which may cause some particles to go uncounted. There are three ways this occurs: First, the particles may be located on the side opposite the one visible in the SEM micrographs of a CNT, rendering them partially or wholly invisible because of shielding by the CNTs. Second, particles may go uncounted due to shielding by the other, larger silver particles above them. Third, the CNTs bearing Ag-NPs may be shielded by CNTs above them. The last shadowing effect may be especially significant because the substrate itself is completely invisible under the CNT “forest,” indicating that a large number of CNTs and therefore the Ag-NPs which adorn them are hidden from view.”

  1. Did author study the cellular uptake?

Answer: We thank the reviewer for this very important study to understand the cellular uptake of the synthesized nanomaterials. In this work, the green technique for preparing SFMWCNTs composites using plant extracts was followed. This process yielded various compounds, the most important of which were Hydroxytyrosol, Tyrosol, and Oleuropein. Olive-leaf extracts contain medicinal properties and combining these extracts with Ag-NPs is often a viable option for enhancing drug delivery; thus, this possibility was employed for in vitro treating cancer cells as a proof of concept. In the future, we might conduct a full cellular uptake and mechanistic aspects of the synthesized nanomaterials.

6.Did author try weight loss experiment nanoparticle due to desorption of organic moieties by TGA?
Answer: We have not conducted this study as we don’t have the instrument aviliable at our lab or research center. In our future studies, we will try to study these suggested studies before going further for animal studies.

  1. Did author see any changes in FT-IR spectrum for olive oil extract and formed nanoparticles?
    Answer: We have conducted FTIR results to our manuscript as following: Section 3.2. and Figure 4. have been added.

3.2. Fourier transform infrared spectroscopy (FTIR) results

FTIR spectroscopy was used to discriminate and identify the biomolecules of olive leaf. FTIR has been utilized by numerous researchers to analyze a variety of materials [8,9]. By looking at FTIR spectra based on stretching or bending vibration of specibonds, FTIR spectroscopy can reveal information on intermolecular interaction. In Figure, the olive leaf's FTIR spectrum is shown. An intense broad band was seen at 3384.58 cm-1, which was caused by polyphenols O-H stretching modes. phenolic compounds and alcohols both contain the hydroxyl (OH) group, which has a broad absorption band at 3310.7 cm-1. The C=C stretch vibration in the aromatic ring and the C=O stretch vibration in polyphenols may have also been responsible for another strong band at 1612.02 and 1386.23 cm-1. The C-H and O-H stretches in alkanes and carboxylic acids have been observed to surface at 2935.37 cm-1, respectively. The C-O bond stretching in amino acids has also resulted in the emergence of a band at 1078.55 cm-1. Previous research has shown that the O-H/N-H, C=C, and C-O-C stretching vibrations are responsible for the FTIR bands that developed at 3384.58 cm-1,1612.02 and 1386.23 cm-1, and 1078.55 cm-1 [8,9].”

Figure 4. FTIR spectroscopy results of the olive-leaf extract and Ag/SFMWCNTs nanocomposite. It can be seen from the FTIR spectrum of Ag NPs that the wavenumber of the OH bending vibration at 3359.29 cm-1 appeared to broaden and decrease, and that the peaks in the (1700-400) cm-1 region almost underwent alteration. 

  1. Author should go through the from the line 110 to 113 and rewrite it. It is bit confusing to understand.
    Answer: Thank you for the comment. The change has been made as suggested.

Reviewer 3 Report

There are some comments that should be addressed before consideration for publication:

1-      The abstract should be rewritten. As far as I know, there is no need to mention to the preparation method in the abstract. Instead, the authors should explain about the aim and results of the study. Explain more about the results not just mentioning the type of tests done and add some numerical results for better understanding.

2-      The introduction should be rewritten. It is too short, does not have enough literature review and background. There is no need to talk about preparation method and results in the introduction. Just mention to the characterization methods and tests done. Read some papers for your references.

3-      The authors should talk about the novelty of their work in the last paragraph of the introduction. What is the novelty of this work?

4-      UV-vis is not an enough and comprehensive test for confirming the formation of AgCNT. FTIR is a better test to understand this. XRD is also fine.

5-      The authors mentioned in the SEM photos that “The former exhibits clusters with almost a circular shape, whereas the composite has a tubular channel structure.”, what does the authors mean by former? The current SEM photo also has some clusters which needs to be explained what they are.

6-       There are some tests that the authors done such as SEM, XRD, UV-vis, EDS, zeta potential and… that should be mentioned in the material and method section how they are performed.

7-      The authors calculated the toxicity of the prepared NCT on cancer cells, but they also calculate the toxicity of these CNT on human cells to see it they have any toxicity or not. How can we sure that they are safe to be used in Human and biomedical applications?

8-      In the discussion section, the authors should talk about their tests and results not other’s. The explanations written in the discussion section should be transferred to the introduction and the authors elaborate and explain about their results here. This section should be modified.

9-      In the discussion section, the authors mentioned that” Polyphenol concentration in olive leaf extract was high, indicating that it has powerful antitumorigenic effects”, but there is no test to confirm that. How do the authors realize that?

Author Response

Dear/ Respected reviewers

Thank you for giving us the opportunity to submit a revised draft of our manuscript titled [Olive Leaf Extracts for a Green Synthesis of Silver-Functionalized Multi-Walled Carbon Nanotubes], manuscript ID: jfb-1944808 to Journal of Functional Biomaterials. I and my co-authors appreciate the time and effort that you and the reviewers have dedicated to providing your valuable and positive feedback on our manuscript. We are grateful to the reviewers for their insightful comments on our paper. We tried as much as possible to respond to most of the enquiries and suggestions provided by the respected reviewers. All changes were made through Microsoft word track changes. Here is our point-by-point response to the reviewers’ comments and concerns followed by references to some responses.

Accept our regards

Reviewer 3:  Comments and Suggestions for Authors

There are some comments that should be addressed before consideration for publication:

1-      The abstract should be rewritten. As far as I know, there is no need to mention to the preparation method in the abstract. Instead, the authors should explain about the aim and results of the study. Explain more about the results not just mentioning the type of tests done and add some numerical results for better understanding.
Answer:  Thank you for the comment. The abstract has been modified as suggested.

2-      The introduction should be rewritten. It is too short, does not have enough literature review and background. There is no need to talk about preparation method and results in the introduction. Just mention to the characterization methods and tests done. Read some papers for your references.
Answer: We thank the reviewer for the comment. We have modified both introduction and discussion sections.

3-      The authors should talk about the novelty of their work in the last paragraph of the introduction. What is the novelty of this work?
Answer: We thank the reviewer for the comment. We have modified the last section of our introduction and added the following paragraph.

In the present study, we aim to combine a comfortable, safe, and cheap method as the water extraction of olive extracts and production of SFMWCNTs, which could be utilized for further investigation on anticancer activities as a single agent or combined with other modalities of treatment. The usage of olive-leaf has the added benefit that nanotechnology processing industries may make use of this plant. It is possible to employ SFMWCNTs nanoparticles produced in this work as anti-cancer agents, and also in the medical field for other diseases. Results indicated the ability of SFMWCNTs to suppress cancer cell expansion and spread was tested using the different cancer cell line. The current approach shows the ability of nanomaterials to enhance cancer growth in-hibition and improves the SFMWCNTs selectivity toward cancer cells.”

4-      UV-vis is not an enough and comprehensive test for confirming the formation of AgCNT. FTIR is a better test to understand this. XRD is also fine.
Answer: We have conducted FTIR results to our manuscript as following: Section 3.2. and Figure 4. have been added.

3.2. Fourier transform infrared spectroscopy (FTIR) results

FTIR spectroscopy was used to discriminate and identify the biomolecules of olive leaf. FTIR has been utilized by numerous researchers to analyze a variety of materials [8,9]. By looking at FTIR spectra based on stretching or bending vibration of specibonds, FTIR spectroscopy can reveal information on intermolecular interaction. In Figure, the olive leaf's FTIR spectrum is shown. An intense broad band was seen at 3384.58 cm-1, which was caused by polyphenols O-H stretching modes. phenolic compounds and alcohols both contain the hydroxyl (OH) group, which has a broad absorption band at 3310.7 cm-1. The C=C stretch vibration in the aromatic ring and the C=O stretch vibration in polyphenols may have also been responsible for another strong band at 1612.02 and 1386.23 cm-1. The C-H and O-H stretches in alkanes and carboxylic acids have been observed to surface at 2935.37 cm-1, respectively. The C-O bond stretching in amino acids has also resulted in the emergence of a band at 1078.55 cm-1. Previous research has shown that the O-H/N-H, C=C, and C-O-C stretching vibrations are responsible for the FTIR bands that developed at 3384.58 cm-1,1612.02 and 1386.23 cm-1, and 1078.55 cm-1 [8,9].”

Figure 4. FTIR spectroscopy results of the olive-leaf extract and Ag/SFMWCNTs nanocomposite. It can be seen from the FTIR spectrum of Ag NPs that the wavenumber of the OH bending vibration at 3359.29 cm-1 appeared to broaden and decrease, and that the peaks in the (1700-400) cm-1 region almost underwent alteration. 

5-      The authors mentioned in the SEM photos that “The former exhibits clusters with almost a circular shape, whereas the composite has a tubular channel structure.”, what does the authors mean by former? The current SEM photo also has some clusters which needs to be explained what they are.
Answer: We thank the reviewer for the comment. We have modified the confusing sentence as suggested. The extracted spherical olive leaf were clustered and SFMWCNT nanocomposite has a tubular-channel structure. “The synthesized materials exhibit clusters with almost a circular shape, whereas the nanocomposite has a tubular-channel structure. Thus, these images indicate amorphous and crystalline structures, respectively. The current findings are compatible with our XRD results.”

6-       There are some tests that the authors done such as SEM, XRD, UV-vis, EDS, zeta potential and… that should be mentioned in the material and method section how they are performed.
Answer: We thank the reviewer for the comment. We have modified materials and methods sections as suggested.

7-      The authors calculated the toxicity of the prepared NCT on cancer cells, but they also calculate the toxicity of these CNT on human cells to see it they have any toxicity or not. How can we sure that they are safe to be used in Human and biomedical applications?

Answer:  We would like to thank the reviewer for the comment CNT have been used in many biomedical applications as the following references indicated [10-12]. Different approaches, including functionalization, as well as their critical functions in targeting distinct intracellular locations and Tumor microenvironments, have been explored to learn more about the development of CNTs as prospective safe drug delivery vehicles in cancer treatment. Furthermore, CNT has seen a lot of recent progress in the field of cancer detection and therapy. As a result, there are a number of reviews and papers that summarize CNTs and their safety for medical applications, such as the paper by Tang, L. and his group [12], who comprehensively introduce the theranostic applications of CNTs against many cancer types from the perspective of various therapeutic targets and emphasize the combination therapeutic modalities based on the physiochemical features of CNTs and compare it with other many reported literatures. In conclusion, CNT was a safe and effective system.

8-      In the discussion section, the authors should talk about their tests and results not other’s. The explanations written in the discussion section should be transferred to the introduction and the authors elaborate and explain about their results here. This section should be modified.
Answer:  We thank the reviewer for the comment. We have modified both introduction and discussion sections.

9-      In the discussion section, the authors mentioned that” Polyphenol concentration in olive leaf extract was high, indicating that it has powerful antitumorigenic effects”, but there is no test to confirm that. How do the authors realize that?
Answer: There are many studies that prove this information and we have cite one of the references. Barbaro, B.; Toietta, G.; Maggio, R.; Arciello, M.; Tarocchi, M.; Galli, A.; Balsano, C. Effects of the olive-derived polyphenol oleuropein on human health. International journal of molecular sciences 2014, 15, 18508-18524.

“Polyphenol concentration in olive leaf extract was high, indicating that it has powerful antitumorigenic effects as previously reported [13].”

Round 2

Reviewer 2 Report

Paper looks good now and can be published

Author Response

Dear Reviewer,

We would like to thank you for your time and effort in reviewing our manuscript. We hope that we answered your questions.

Best Regards 

Reviewer 3 Report

The authors addressed almost most of the comments but one comment left unaddressed:

 There are some tests that the authors done such as SEM, XRD, UV-vis, EDS, zeta potential and… that should be mentioned in the material and method section how they are performed. The authors mentioned that " We have modified materials and methods sections as suggested." But I could not find it in the material and methods section

Author Response

Dear Reviewer,

Thank you for giving us the opportunity to resubmit a revised draft of our manuscript titled [Olive Leaf Extracts for a Green Synthesis of Silver-Functionalized Multi-Walled Carbon Nanotubes], manuscript ID: jfb-1944808 to Journal of Functional Biomaterials. I and my co-authors appreciate the time and effort that you and the reviewers have dedicated to providing your valuable and positive feedback on our manuscript. We are grateful to the reviewers for their insightful comments on our paper. We tried as much as possible to respond to most of the enquiries and suggestions provided by the respected reviewers. All changes were made through Microsoft word track changes. Here is our answer about the reviewer's comment.

Comment:

There are some tests that the authors done such as SEM, XRD, UV-vis, EDS, zeta potential and… that should be mentioned in the material and method section how they are performed. The authors mentioned that " We have modified materials and methods sections as suggested." But I could not find it in the material and methods section

Answer: We thank the reviewer for the comment. We have modified materials and methods sections as suggested in section 2.3 (2.3.1 and 2.3.2).